# Relationship of Societal Adaptation with Vaccine Worries among Healthcare Workers during the COVID-19 Pandemic: The Mediating Effects of Posttraumatic Stress Disorder

**DOI:** 10.3390/ijerph19159498

**Published:** 2022-08-02

**Authors:** Kuan-Ying Hsieh, Dian-Jeng Li, Frank Huang-Chih Chou, Su-Ting Hsu, Hui-Ching Wu, Li-Shiu Chou, Pei-Jhen Wu, Guei-Ging Lin, Wei-Jen Chen, Chin-Lien Liu, Joh-Jong Huang

**Affiliations:** 1Kaohsiung Municipal Kai-Syuan Psychiatric Hospital, Kaohsiung 80276, Taiwan; isanrra@kcg.gov.tw (K.-Y.H.); u108800004@kmu.edu.tw (D.-J.L.); hsust@kcg.gov.tw (S.-T.H.); lschou@kcg.gov.tw (L.-S.C.); f0168@kcg.gov.tw (P.-J.W.); t1921118@kcg.gov.tw (G.-G.L.); chenweijen@kcg.gov.tw (W.-J.C.); amelie12@kcg.gov.tw (C.-L.L.); 2Graduate Institute of Medicine, College of Medicine, Kaohsiung Medical University, Kaohsiung 80708, Taiwan; 3Department of Nursing, Meiho University, Pingtung 91200, Taiwan; 4Graduate Institute of Counseling Psychology and Rehabilitation Counseling, National Kaohsiung Normal University, Kaohsiung 80201, Taiwan; 5Department of Social Work, National Taiwan University, Taipei 10617, Taiwan; hchingwu@ntu.edu.tw; 6Department of Medical Humanities and Education, Kaohsiung Medical University, Kaohsiung 80708, Taiwan; 7Department of Family Medicine, Kaohsiung Medical University Hospital, Kaohsiung 80756, Taiwan

**Keywords:** coronavirus disease 2019 (COVID-19), posttraumatic stress disorder, healthcare workers, vaccine hesitancy, societal adaptation

## Abstract

Vaccine hesitancy has become a major public health problem among healthcare workers (HCWs) in this coronavirus disease 2019 (COVID-19) pandemic. The aim of this study was to examine the relationship between societal adaptation and vaccine worries and the mediating effects of posttraumatic stress disorder (PTSD) indicators in HCWs. A total of 435 HCWs (327 women and 108 men) were recruited. Their levels of societal adaptation were evaluated using the Societal Influences Survey Questionnaire (SISQ). Their severity and frequency of PTSD symptoms were examined using the Disaster-Related Psychological Screening Test (DRPST). The severity of vaccine worries was assessed using the Vaccination Attitudes Examination (VAX) Scale. The relationships among societal adaptation, PTSD, and vaccine worries were examined using structural equation modeling. The severity of societal adaptation was positively associated with both the severity of PTSD and the severity of vaccine worries. In addition, the severity of PTSD indicators was positively associated with the severity of vaccine worries. These results demonstrated that the severity of societal adaptation was related to the severity of vaccine worries, either directly or indirectly. The indirect relationship was mediated by the severity of PTSD. Societal adaptation and PTSD should be taken into consideration by the community of professionals working on vaccine hesitancy. Early detection and intervention of PTSD should be the objectives for programs aiming to lower vaccine hesitancy among HCWs.

## 1. Introduction

### 1.1. Societal Adaptation during COVID-19 Pandemic

Coronavirus disease 2019 (COVID-19) has been spread worldwide since the end of 2019, and deeply affected people′s daily lives in health and social-economic status [1]. Therefore, COVID-19 has become a biological disaster [2,3].

Societal adaptation consists of the behavior changes to prevent COVID-19 infection [4]. A study in German reported protective behaviors as washing hands more often and longer, avoiding busy places or reduced personal meetings and contacts, keeping social distance, using disinfectants, and school/work adaption [5]. It is still unclear whether individuals are aware of infection risk and adapt their vaccinated behaviors during the COVID-19 pandemic.

### 1.2. Vaccine Hesitancy in Healthcare Workers (HCWs)

By achieving herd immunity, vaccination is the most hopeful way of controlling COVID-19 since December 2020 [6,7,8]. However, the achievement of herd immunity is a challenge, because the vaccine attitude in the general population was complex and attributed by many factors [9,10].

Vaccine hesitancy has increased in the past 20 years due to the global anti-vaccination message transmission through social media and Internet [11]. Although the politicization of the COVID-19 virus and the speedy vaccine development likely increase vaccine hesitancy, researches on factors related to vaccination beliefs are needed [12].

Healthcare workers (HCWs), not only at risk of infection [13], but also provided vaccination information to patients, are the key to the success of COVID-19 vaccination [14]. HCWs has high priority in COVID-19 vaccination in many countries. However, vaccine hesitancy, including vaccination refusal or delay, in HCWs was noted [15]. Clarifying factors related to HCWs′ attitude toward COVID-19 vaccines can provide new solutions for vaccine hesitancy.

### 1.3. Psychological Impact of HCWs

HCWs in Taiwan are vulnerable in PTSD during the COVID-19 pandemic due to previous experience of SARS, infection risk, work overload, and emotional burden [16,17,18]. The most common psychiatric disorders of COVID-19 on HCWs were post-traumatic stress disorder (PTSD), depression and anxiety in several studies [19,20,21,22,23,24,25,26,27,28,29,30,31,32,33,34,35,36,37,38,39,40,41,42,43,44,45,46]. A study in China revealed that the prevalence rates of anxiety, depression and insomnia were 5.9% in doctors and 35% in nurses. Furthermore, the risk factors were female, low social support, contact with confirmed or suspected cases and working on the clinical front-line in China [47]. Overworked and under-resourced HCWs could refuse to work by infection risk. Therefore, shortages of HCWs could happen, which may further exacerbate the overwork problems [48]. Hence, psychological impacts of HCWs shouldn′t be omitted, even relatively smaller cases in Taiwan than other countries.

Surprisingly, few studies have examined the relationship between mental health and vaccination hesitancy. A study reported that individuals with depression or anxiety history are more willing to get influenza vaccinations compared to those without [49]. In contrast, a study reported the relationship between children with low immunization rates and their mothers with a depression history [50], although null results have also been reported in another study [51]. No known studies have investigated how PTSD relates to vaccine hesitancy in HCWs.

### 1.4. Aims of This Study

No study has examined the relationship between societal adaption of COVID-19, PTSD, and vaccine hesitancy. This study examined the mediating effects of PTSD on the relationship between societal adaptation and vaccine worries in HCWs. We hypothesized that PTSD is positively associated with vaccine worries and that PTSD mediate the relationship between societal adaptation and vaccine worries.

## 2. Materials and Methods

### 2.1. Participants and Ethics

This is our second-year follow up study by our group, and the first year outcome has been published elsewhere [4,52,53,54]. We recruited participants through printed advertisements posted in the public area in the Kaohsiung Municipal Kai-Syuan Psychiatric Hospital in Taiwan, from 30 May 2021 to 30 June 2021. This study is a cross-sectional survey with paper-and-pencil questionnaires, and research assistants individually explained the procedures to the participants in order to complete the research questionnaires. The inclusion criteria were: (1) healthcare team members who were willing to participate, (2) could understand the objective of the study and follow the instructions from research assistants, (3) were aged between 20 to 80 years, and (4) signed informed consent before filling in the questionnaire. Data with missing values or from those who could not complete the questionnaire were excluded.

This study was approved by the institutional review board of Kai-Syuan Psychiatric Hospital (KSPH-2020-03). This study was registered at Clinicaltrials.gov (Identifier number: NCT04389476).

### 2.2. Measures

#### 2.2.1. Post-Traumatic Stress Disorder Scale from the Disaster-Related Psychological Screening Test (DRPST)

We used the PTSD scale of DRPST in this study to assess the level of PTSD severity and frequency. DRPST has been tested with acceptable reliability and validity in screening for PTSD after a disaster [55,56]. Eight items were used to estimate the status of PTSD severity and frequency, including re-experiencing, avoidance, numbing and arousal which had persisted for more than 2 weeks in the preceding 1 month. Each item was rated on a 5-point Likert scale, with scores ranging from 0 (never/no) to 5 (everyday/profound). Higher total scores of the three items indicated higher levels of PTSD frequency and severity. Comparing with a relatively low total score, a relatively high total score indicates a higher level of trauma-related symptoms. The Cronbach′s α of PTSD severity scale and PTSD frequency scale of DRPST in this study was 0.86 and 0.84, respectively, which was in an acceptable range [57].

#### 2.2.2. Social Adaptation Category from Societal Influences Survey Questionnaire (SISQ)

The Societal Influences Survey Questionnaire (SISQ) was developed to measure the psychological and social impact of the COVID-19 pandemic on individuals, and it was tested as having acceptable validity and reliability [58]. We used the social adaptation category of societal influences survey questionnaire to evaluate the level of social adaptation regarding to COVID-19. Four items were used to estimate the status of adaptive behaviors related to COVID-19, including awareness of progress of the pandemic overseas in the preceding 1 month. Each question was composed of a 4-point scale, with scores ranging from 1 (never) to 4 (often). A relatively high total score of social adaptation domain indicates a higher level of social desirability than a relatively low total score. The Cronbach’s α of social adaptation domain of SISQ in the present study was 0.48, which was in an acceptable range [57].

#### 2.2.3. Worries about Unforeseen Future Effects Domain from Vaccination Attitudes Examination (VAX) Scale

We used the worries about unforeseen future effects domain of the Vaccination Attitudes Examination (VAX) scale to evaluate the participants’ level of worries about COVID-19 vaccine in this study. VAX has been reported with good validity and reliability [59]. Three items were used to estimate the status of worries related to COVID-19 vaccine in the preceding 1 month. Each question was composed of a 6-point scale, with scores ranging from 1 (not at all) to 6 (extreme). A relatively high total score of the worries about unforeseen future effects domain indicates higher level of vaccine worries than a relatively low total score. The Cronbach’s α of worries about unforeseen future effects domain in this study was 0.64, which was in an acceptable range [57].

#### 2.2.4. Demographic Characters

Other demographic information was recorded as continuous variables, including the participants’ age and their education level. Categorical variables including gender, occupational status, and marital status, were also recorded.

### 2.3. Statistical Analysis

The hypothesized model for the relationship among societal adaption, vaccine worries, and PTSD is presented in Figure 1. We hypothesized that societal adaptation is positively associated with PTSD and vaccine worries. Furthermore, PTSD mediates the relationship between societal adaptation and vaccine worries.

We used structural equation model (SEM) to exam parameters, model adequacy, and evaluate the agreement between the data and the model [60]. We used the Amos 18.0 software (IBM SPSS, Armonk, NY, USA), adapting the maximum likelihood method, to calculate goodness-of-fit index (GFI), non-normed fit index (NNFI), incremental fit index (IFI), comparative fit index (CFI), root mean square error of approximation (RMSEA), and standardized root mean square residual (SRMR) [61]. According to the goodness-of-fit requirement, NNFI, GFI, IFI, and CFI need to be higher than 0.9; the values of RMSEA and SRMR between 0.05–0.09 are acceptable [61]. We used Sobel test to test the mediating effect of PTSD on the relationship between societal adaptation and vaccine worries. A two-tailed *p* value of less than 0.05 was considered statistically significant.

## 3. Results

### Summary of Demographic Analysis and Scores of the Questionnaires

There were 560 individuals recruited in this study, but 125 who did not complete the copy of the questionnaire. Four hundred and thirty-five participants (327 females and 108 males) were analyzed (Table 1). There were no significant differences in sex or age between those who completed the coy of questionnaire and those who did not.

The correlation matrix of measured variables are shown in Table 2. The results showed the PTSD, social adaptation and vaccine worries were all significant correlated to each other.

The goodness-of-fit indices of SEM for the hypothesized model on the relationship among societal adaption, vaccine worries, and PTSD are presented in Table 3. The estimated coefficients of paths in our hypothesized model are showed in Figure 2. The goodness-of-fit indices for the hypothesized model were all acceptable. The result of the Sobel test certificated the mediating effect of PTSD on the relationship between societal adaptation and vaccine worries (Z = 2.235, *p* < 0.05).

Moreover, we found that all hypothesized paths were significant. The level of societal adaptation was positively associated with the severity of vaccine worries and PTSD. Additionally, the severity and frequency of PTSD symptoms were positively associated with the severity of vaccine worries. Furthermore, the severity of societal adaptation was related straightforwardly to the severity of vaccine worries and related concomitantly to the severity of vaccine worries through increasing the severity and frequency of PTSD symptoms.

## 4. Discussion

Our study was the first study to examine the relationship between societal adaptation, vaccine worries, and PTSD among HCWs. The results of this study demonstrated that societal adaptation was directly related to vaccine worries and indirectly related to vaccine worries by the mediation of PTSD.

### 4.1. Societal Adaptation and Vaccine Worries

The relationship between societal adaptation and vaccine attitude has been complicated. People took societal adaptive behaviors based on risk perception. Risk perception is defined as personal judgement regarding an event [62]. Risk interpretation is influenced by cognition, affective reactions [63], and contextual factors [64]. A study carried out showed that people who sensed a greater risk of COVID-19 infection were more likely to have adaptive behaviors such as washing their hands and maintaining social distancing [65]; another study reported that risk perception was reported to be positively correlated with depression in patients with COVID-19 [66]. A meta-analysis, examined the relationship between risk appraisals, intentions and behaviors, revealed that heightening risk appraisals changes intentions and behavior [67]. Therefore, it is reasonable that when people perceive a heightened risk of COVID-19, they are more likely to take adaptive behaviors, including vaccination [68]. However, in our study, we found societal adaption was positive related to vaccine worries. This phenomena was also reported in several studies. A study in the U.S showed a decrease in COVID-19 vaccine intentions during March 2020 to August 2020 [69]. Another study in the United States explored disease risk awareness and behavior adaption [70]. Participants showed increasing awareness of risk and reported taking protective behaviors with increasing frequency, comparing with general population.

The motivations of vaccination were diverse. A study examined the differences in intention to receive a COVID-19 vaccination between HCWs and the general population by Protection Motivation Theory (PMT) in Taiwan [71]. They found that HCWs were more motivated in taking COVID-19 vaccination than the general population. Responses of COVID-19 vaccination such as efficacy and knowledge were positively associated with vaccination motivation in both HCWs and the general population, but the awareness of vulnerability, severity, and response cost of COVID-19 vaccination was positively associated with motivation in the general population, but not in HCWs. One of the explanation is that HCWs may have different coping strategies toward COVID-19 from general population [72].

There are several possible explanations for the positive association between societal adaption and vaccine worries. First, HCWs with higher levels of adaptive behaviors are confident with their infection control strategies, and therefore disregard the need for vaccination. Second, HCWs with full medical training worry about rare but severe adverse effects of the vaccine [73], especially in a rapid development process [74].

### 4.2. Societal Adaptation and PTSD

HCWs may have severe and lasting emotional stress in large-scale disease epidemics [75,76,77]. In terms of the psychological impact of COVID-19, HCWs are vulnerable for infection risk, workload and transmission to their families [78]. At the beginning of the COVID-19 pandemic, the prevalence of mental health problems was 40% in China HCWs [22,45]. Two-weeks after Wuhan lockdown, the prevalence of female HCWs with depression, anxiety and stress symptoms were ranging from 10–30% [79]. A Spanish study revealed HCWs had more stress-related symptoms than non-HCWs [20]. A Chinese study reported that frontline nurses were more traumatized than non-frontline nurses [80]. Another study reported that HCWs were more fearful compared with to non-clinical colleagues [81]. Above findings suggested that HCWs suffered higher levels of psychological symptoms, including anxiety, depression and acute stress symptoms, than the general population.

Individuals with good societal adaptation may have good coping strategies toward stress. There are several different coping strategies, including acceptance, resilience, active coping and positive framing [82]. Furthermore, doctors prefer plan, yet nurses use behavioral disengagement [82]. A study focusing on HCWs in Taiwan showed that female, older, more education years, married, regular intake, and higher PTSD frequency suffered more societal influence [4]. Exercise was a commonly used coping strategy in American HCWs, followed by online self-guided counselling with an individual therapist [32]. However, HCWs who adapted well at first with their own coping strategies may be exhausted, helpless, hopeless, have negative thoughts of the future, which then increases the risk of PTSD in this long-lasting COVID-19 pandemic.

### 4.3. PTSD and Vaccine Worries

We found that the severity and frequency of PTSD were positively correlated with vaccine worries in our SEM model (Figure 1). There are conceptual reasons to believe that PTSD is relevant to vaccine hesitancy. Persisting negative beliefs toward oneself, others, or the world is one of the PTSD criteria defined by the fifth edition of Diagnostic and Statistical Manual of Mental Disorders (DSM-5) [83]. Individuals who experience PTSD develop mistrust of others and institutions, a negative view of the world [84,85,86], and a lack of trust in healthcare providers [87]. The impact of distrust could persist years after the event, including in daily life and health behaviors [88], as well as reducing use of recommended healthcare services by traumatized individuals [89,90]. Distrust is pivotal for vaccine behaviors [91]. Vaccine decisions rely on trust in various aspects, including the efficacy (pharmaceutical companies), safety (the government), and delivery (medical providers) [12]. Therefore, aggressively detection and treatment PTSD in HCWs is important.

Our study highlight the importance of trauma related problems when addressing HCWs′ vaccine hesitancy. First, we suggested that early detection of mental health problems [39,46], especially PTSD, in HCWs by authorities or hospital administrators, as well as the extent and sources of stress among HCWs [75]. Moreover, hospital support systems should promote the psychological wellbeing of HCWs [92] by additional psychosocial support [93] with effective strategies [94] and adequate care [95]. Previous studies had empathized the importance of collaboration, training and education [77,82,96,97,98]. In addition, health authorities should provide support and training, counselling hotlines and offer reimbursements to HCWs [93,99]. Second, HCWs need to be informed about the COVID-19 vaccine development process and frame health messaging [100].

### 4.4. Limitations

The current study has several limitations. First, our data were self-reported and shared-method variance should be considered. Second, a single-center study may limit the generalizability. The diversity source, participants included doctors, nurses, occupational therapists, clinical psychologists, pharmacists, medical technologist, medical radiation technologist and physical therapists, may make the results complicated. Third, we did not examine the effect of gender in this study. Previous studies have suggested that females were more susceptible to develop symptoms of PTSD than males [101,102]. Female nurses taking care of COVID-19 patients had higher mental health risks than others [34,41,103]. In comparison with men, women express stronger emotions, more negative views of health, and then more psychological problems [104]. Women also exhibit differential neurobiological responses when exposed to stressors than men, and therefore are at risk of PTSD [105,106]. Fourth, we did not ascertain the participants′ physical diagnoses in this study. Patients with chronic diseases suffered from compromised immunity and a higher risk of mortality, and therefore were anxious toward COVID-19 infection [107,108,109,110]. The role of physical illness in the pathogenesis of PTSD was postulated that chronic physical illnesses influenced patients′ self-efficacy, and then reduced their recovery potential [111]. Fifth, the cross-sectional research design limited the ability to draw a causal relationship between societal adaption, vaccine worries and PTSD. Further longitudinal study can be helpful.

## 5. Conclusions

We found that societal adaptation was directly related to vaccine worries and indirectly related to vaccine worries by the mediation of PTSD. Therefore, societal adaptation and PTSD should be considered in the work of vaccine hesitancy. We suggested early detection and intervention for PTSD to reduce HCWs’ vaccine hesitancy.

## Figures and Tables

**Figure 1 ijerph-19-09498-f001:**
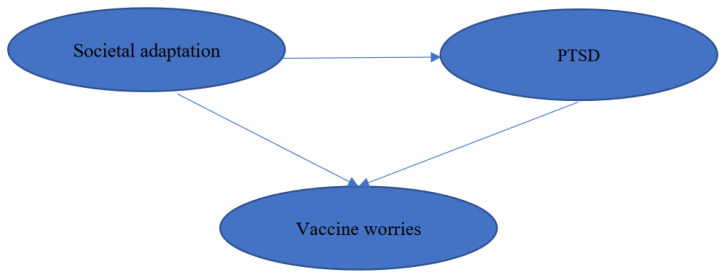
Hypothesized model of the associations among societal adaptation, PTSD and vaccine worries. PTSD, posttraumatic stress disorder.

**Figure 2 ijerph-19-09498-f002:**
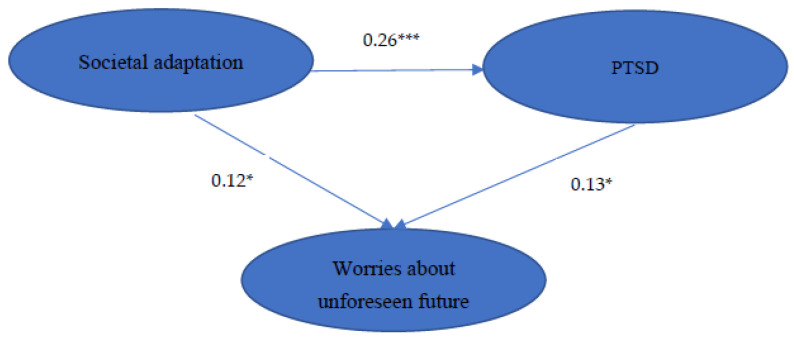
The conceptual model showing interrelationships between societal adaptation, PTSD and vaccine worries. *: *p* < 0.05; ***: *p* < 0.001. PTSD, post-traumatic stress disorder.

**Table 1 ijerph-19-09498-t001:** The demographic data and questionnaire′s among participant (N = 435).

	Participants (N = 435) n (%)
Female	327 (75.2)
Age (years), mean ± SD	38.3 ± 10.6
Education (years), mean ± SD	16.5 ± 2.8
With spouse	197 (45.3)
Doctors	40 (9.2)
Nurses	250 (57.5)
Personal Care Attendant	54 (12.4)
Others	91 (20.9)
DRPST PTSD severity, mean ± SD	1.7 ± 2.1
DRPST PTSD frequency, mean ± SD	4.6 ± 4.4
Social adaptation, mean ± SD	13.2 ± 2.7
Vaccine worries, mean ± SD	12.3 ± 2.5

SD, standard deviation. DRPST, disaster-related psychological screening test. PTSD, post-traumatic stress disorder.

**Table 2 ijerph-19-09498-t002:** The correlation matrix of measured variables.

	1	2	3	4
1. PTSD severity	1	0.83 ***	0.21 ***	0.16 **
2. PTSD frequency		1	0.26 ***	0.15 **
3. Social adaptation			1	0.15 **
4. Vaccine worries				1

** *p* < 0.01, *** *p* < 0.001; SD, standard deviation; PTSD, post-traumatic stress disorder.

**Table 3 ijerph-19-09498-t003:** The Goodness-of-Fit index of structural equation modeling for the hypothesized model.

Type	Goodness of Fit Index	The Full Model
Absolute fit indices	χ^2^	1.533
	df	1
	RMSEA	0.035 (*p* < 0.05)
	GFI	0.998 (*p* > 0.09)
Relative fit indices	NFI	0.997 (*p* > 0.09)
	IFI	0.999 (*p* > 0.09)
	CFI	0.999 (*p* > 0.09)
	SRMR	0.009 (*p* < 0.05)

χ^2^: chi-square; RMSEA: Root Mean Square Error of Approximation; GFI: Goodness-of-Fit Index; NFI: Non-normed-Fit; Index; IFI: Incremental Fit Index; CFI: Comparative Fit Index; SRMR: Standardized Root Mean Square Residual. df: degree of freedom.

## Data Availability

Not applicable.

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
