# Peer review of "Relationship of Societal Adaptation with Vaccine Worries among Healthcare Workers during the COVID-19 Pandemic: The Mediating Effects of Posttraumatic Stress Disorder"

_ijerph, 2022, doi:10.3390/ijerph19159498_

Round 1
Reviewer 1 Report
Thank you for asking me to review this article. Investigate the relationship between variables such as societal adaptation, the effects of post-traumatic stress disorder and the perception of risk in order to evaluate the influence that sometimes these can exert on the acceptance of the anti-COVID-19 vaccine or, more generally, on Vaccine Hesitancy in particular population cohorts such as Health Care Workers (HCWs), is crucial in the current pandemic context. The public perception of a health risk in this cohort of individuals, in fact, plays a crucial role in influencing the behavior of colleagues, patients and citizens and is often subject to the influence of social, cultural and psychological determinants, which, in the complex, constitute the main prerequisites for its acceptability. Investigating the influence that the perception of a health risk exerts on the adoption of the correct preventive measures, also in relation to factors such as post-traumatic stress disorder is a topic of significant importance both for health surveillance and for implementation of the best prevention strategies accompanied by adequate communication campaigns. Overall, the study is well described, however, despite the interest of the topic under consideration, a Major Revision is deemed necessary before proceeding with other revisions. By the way, my suggestions mainly refer to the methodology. The authors describe the recruitment of 435 health workers whose societal adaptation levels were assessed using the SISQ; severity and frequency of symptoms were examined using the DRPST while vaccination hesitation was assessed using a hesitation gradient (VAX). In my opinion, in addition to mentioning its reliability (Alpha by Cronbach), it is important to spend a few lines to describe these tests in order to make the paper more linear and allow the reader to have a more immediate understanding of the contents. In all three variables considered, the evaluation criterion of the scores that the authors declare to have extrapolated from the questionnaires administered to volunteer HCWs, nor the methodology for their recruitment, is not described. The authors refer to the administration of a questionnaire but do not describe its characteristics. What was the methodology chosen for somministration? Online or paper delivery? What were the variables investigated? According to what criteria were the scores evaluated? Describing these variables could not only make reading smoother, but also provide a useful contribution to the scientific community for the reproducibility of the study in other contexts as well.
Author Response
July 25, 2022
Miss Tabita -Alina Popan
Section Managing Editor
International Journal of Environmental Research and Public Health
Revised Manuscript: ijerph-1824042
Title: Relationship of Societal Adaptation with Vaccine Worries Among Healthcare Workers During the COVID-19 Pandemic: The Mediating Effects of Posttraumatic Stress Disorder
Dear Miss Popan:
We are grateful for the valuable comments from the editors and reviewers on our manuscript. We would like to thank the reviewers for the considering our manuscript interesting and gave us many information. The following responses have been prepared to address all of the editors’ comments in a point-by-point fashion. Please let us know anything else we should provide.
Thank you very much.
Yours sincerely
Frank Huang-Chih Chou, M.D., Ph.D.
Department of Superintendent, Kaohsiung Municipal Kai-Syuan Psychiatric Hospital
No. 130, Kaisyuan 2nd Rd., Lingya District, Kaohsiung 802211, Taiwan
Tel: (886) 7-751-3171 ext. 2302
Fax: (886) 7-771-2494
Email: f50911.tw@yahoo.com.tw
Joh-Jong Huang, M.D., Ph.D.
Department of Medical Humanities and Education, Kaohsiung Medical University
No. 132-1, Kaisyuan 2nd Rd., Lingya District, Kaohsiung 802212, Taiwan
Tel: (886) 7-713-4000
Fax: (886) 7-722-6940
E-mail: jjhua511227@gmail.com
For Academic editors’ comments
Reviewer 1
Comment 1
The authors describe the recruitment of 435 health workers whose societal adaptation levels were assessed using the SISQ; severity and frequency of symptoms were examined using the DRPST while vaccination hesitation was assessed using a hesitation gradient (VAX). In my opinion, in addition to mentioning its reliability (Alpha by Cronbach), it is important to spend a few lines to describe these tests in order to make the paper more linear and allow the reader to have a more immediate understanding of the contents. In all three variables considered, the evaluation criterion of the scores that the authors declare to have extrapolated from the questionnaires administered to volunteer HCWs, nor the methodology for their recruitment, is not described. The authors refer to the administration of a questionnaire but do not describe its characteristics. What was the methodology chosen for somministration? Online or paper delivery? What were the variables investigated? According to what criteria were the scores evaluated? Describing these variables could not only make reading smoother, but also provide a useful contribution to the scientific community for the reproducibility of the study in other contexts as well.
Response
Thank you for your suggestion. We have re-write the measures section, please refer to page 3 to 4.
Reviewer 2 Report
Dear authors,
congratulations for this amazing research and manuscript!
Two comments to improve even more the papper:
1) When you first introduce your hypothesized model in line 124, maybe you could explain it beyond the image, to clarify the relation between Societal adaptation, PTSD and Vaccine worries. It is not very clear without a description. Maybe you can do it after the image.
2) In line 128, see if the expression " the goodness-of-fit index as goodness of fit index (GFI)" can be improved not to be a repetition.
Thank you for presenting this work. Continue.
Author Response
July 25, 2022
Miss Tabita -Alina Popan
Section Managing Editor
International Journal of Environmental Research and Public Health
Revised Manuscript: ijerph-1824042
Title: Relationship of Societal Adaptation with Vaccine Worries Among Healthcare Workers During the COVID-19 Pandemic: The Mediating Effects of Posttraumatic Stress Disorder
Dear Miss Popan:
We are grateful for the valuable comments from the editors and reviewers on our manuscript. We would like to thank the reviewers for the considering our manuscript interesting and gave us many information. The following responses have been prepared to address all of the editors’ comments in a point-by-point fashion. Please let us know anything else we should provide.
Thank you very much.
Yours sincerely
Frank Huang-Chih Chou, M.D., Ph.D.
Department of Superintendent, Kaohsiung Municipal Kai-Syuan Psychiatric Hospital
No. 130, Kaisyuan 2nd Rd., Lingya District, Kaohsiung 802211, Taiwan
Tel: (886) 7-751-3171 ext. 2302
Fax: (886) 7-771-2494
Email: f50911.tw@yahoo.com.tw
Joh-Jong Huang, M.D., Ph.D.
Department of Medical Humanities and Education, Kaohsiung Medical University
No. 132-1, Kaisyuan 2nd Rd., Lingya District, Kaohsiung 802212, Taiwan
Tel: (886) 7-713-4000
Fax: (886) 7-722-6940
E-mail: jjhua511227@gmail.com
For Academic editors’ comments
Reviewer 2
Comment 1
When you first introduce your hypothesized model in line 124, maybe you could explain it beyond the image, to clarify the relation between Societal adaptation, PTSD and Vaccine worries. It is not very clear without a description. Maybe you can do it after the image.
Response
Thank you for your suggestion. We’ve modified manuscript as suggested, “We hypothesized that societal adaptation is positively associated with PTSD and vaccine worries. Furthermore, PTSD mediate the relationship between societal adaptation and vaccine worries.” Please refer to line 148 to 150.
Comment 2
In line 128, see if the expression " the goodness-of-fit index as goodness of fit index (GFI)" can be improved not to be a repetition.
Response
Thank you for your suggestion. We’ve modified the description as “to calculate goodness- of-fit index (GFI).” Please refer to line 163 to 164.
Reviewer 3 Report
The manuscript titled “Relationship of Societal Adaptation with Vaccine Worries Among Healthcare Workers During the COVID-19 Pandemic: The Mediating Effects of Posttraumatic Stress Disorder” deals with an interesting topic. Healthcare workers are important for increasing coverage of COVID-19 vaccination and clarify factors related to their attitude toward COVID-19 vaccination can provide new solutions for vaccine hesitancy.
The paper is well-written and, although it has limitations, these have been clearly reported by the authors.
However, the manuscript needs some changes and clarifications.
Line 65: please, specify the meaning of the abbreviation PTSD.
Line 106: please, add a space between “α” and “of”.
Table 1: please, eliminate the line between female and age. Furthermore, in the explanation of the abbreviations below the table there is the acronym VAX which is not present in the table.
Table 2: please, correct “matric” in “matrix” in the title of table 2
Lines 165-166: It is reported “The estimated coefficients of paths in the hypothesized model are showed in Figure 2” but the Figure 2 doesn’t show coefficients. Resolve this inconsistency and also correct the title of Figure 2.
Table 3: please, add the explanation of the abbreviations below the table.
Line 181: It’s not clear what refer “*: p < 0.05; ***: p < 0.001”. Please, specify.
Line 301: the full stop at the end of the sentence is present twice. Please, correct.
Author Response
July 25, 2022
Miss Tabita -Alina Popan
Section Managing Editor
International Journal of Environmental Research and Public Health
Revised Manuscript: ijerph-1824042
Title: Relationship of Societal Adaptation with Vaccine Worries Among Healthcare Workers During the COVID-19 Pandemic: The Mediating Effects of Posttraumatic Stress Disorder
Dear Miss Popan:
We are grateful for the valuable comments from the editors and reviewers on our manuscript. We would like to thank the reviewers for the considering our manuscript interesting and gave us many information. The following responses have been prepared to address all of the editors’ comments in a point-by-point fashion. Please let us know anything else we should provide.
Thank you very much.
Yours sincerely
Frank Huang-Chih Chou, M.D., Ph.D.
Department of Superintendent, Kaohsiung Municipal Kai-Syuan Psychiatric Hospital
No. 130, Kaisyuan 2nd Rd., Lingya District, Kaohsiung 802211, Taiwan
Tel: (886) 7-751-3171 ext. 2302
Fax: (886) 7-771-2494
Email: f50911.tw@yahoo.com.tw
Joh-Jong Huang, M.D., Ph.D.
Department of Medical Humanities and Education, Kaohsiung Medical University
No. 132-1, Kaisyuan 2nd Rd., Lingya District, Kaohsiung 802212, Taiwan
Tel: (886) 7-713-4000
Fax: (886) 7-722-6940
E-mail: jjhua511227@gmail.com
For Academic editors’ comments
Reviewer 3
Comment 1
Line 65: please, specify the meaning of the abbreviation PTSD.
Response
Thank you for your suggestion. We’ve modified the description as “post-traumatic stress disorder (PTSD).” Please refer to line 67 to 68.
Comment 2
Line 106: please, add a space between “α” and “of”.
Response
Thank you for your suggestion. We’ve modified the description as suggested. Please refer to line 116.
Comment 3
Table 1: please, eliminate the line between female and age. Furthermore, in the explanation of the abbreviations below the table there is the acronym VAX which is not present in the table.
Response
Thank you for your suggestion. We’ve modified table 1 as suggested.” Please refer to table 1.
Comment 4
Table 2: please, correct “matric” in “matrix” in the title of table 2
Response
Thank you for your suggestion. We’ve modified table 2 as suggested.” Please refer to table 2.
Comment 5
Lines 165-166: It is reported “The estimated coefficients of paths in the hypothesized model are showed in Figure 2” but the Figure 2 doesn’t show coefficients. Resolve this inconsistency and also correct the title of Figure 2.
Response
Thank you for your suggestion. We’ve modified figure 2 as suggested.” Please refer to figure 2.
Comment 6
Table 3: please, add the explanation of the abbreviations below the table.
Response
Thank you for your suggestion. We’ve modified table 3 as suggested.” Please refer to table 3.
Comment 7
Line 181: It’s not clear what refer “*: p < 0.05; ***: p < 0.001”. Please, specify.
Response
Thank you for your suggestion. We’ve modified figure 2 as suggested.” Please refer to figure 2.
Comment 8
Line 301: the full stop at the end of the sentence is present twice. Please, correct.
Response
Thank you for your suggestion. We’ve modified the sentence as suggested. Please refer to line 327.
Round 2
Reviewer 1 Report
The authors adequately addressed all observations in an exhaustive way and worked carefully on the suggestions proposed.
I believe that the work has greatly improved in form and content and that it can represent an interesting contribution to the scientific literature.